# Evaluation of Four Fully Integrated Molecular Assays for the Detection of Respiratory Viruses during the Co-Circulation of SARS-CoV-2, Influenza and RSV

**DOI:** 10.3390/jcm11143942

**Published:** 2022-07-06

**Authors:** Eric Farfour, Thomas Yung, Robin Baudoin, Marc Vasse

**Affiliations:** 1Service de Biologie Clinique, Hôpital Foch, 92150 Suresnes, France; t.yung@hopital-foch.com (T.Y.); m.vasse@hopital-foch.com (M.V.); 2Service d’Otho-Rhino-Laryngologie, Hôpital Foch, 92150 Suresnes, France; r.baudoin@hopital-foch.com

**Keywords:** Idylla, COVID-19, ID NOW, Flu, influenza, SARS-CoV-2, respiratory syncytial virus, Abbott, Biocartis, RSV

## Abstract

**Background**: The clinical presentation of viral respiratory infections is unspecific. We assessed the performances of two new RT-PCR, the Idylla™ SARS-CoV-2 and the Idylla™ SARS-CoV2/Flu/RSV, and two isothermal amplification assays, the ID NOW COVID and the ID NOW influenza A & B 2. **Methods**: The study was conducted in two parts: (i) the Idylla™ assays were assessed using a collection of nasopharyngeal swabs which were positive for various respiratory viruses. (ii) The performances of the four assays were assessed prospectively: all of the symptomatic patients admitted to the emergency department from 10 to 21 December were enrolled. **Results:** (i) All of the SARS-CoV-2 false negatives with the Idylla™ assays had a Ct value greater than 30 with the reference RT-PCR. No cross-reactivity was identified. (ii) Overall, 218 patients were enrolled. The respective prevalences of SARS-CoV-2, influenza A, and RSV were 19.8%, 4.8%, and 3.2%. All of the assays were 100% specific. The sensitivity of SARS-CoV-2 detection was 97.7%, 82.5%, and 86.3% for the Idylla™ SARS-CoV2, the Idylla™ SARS-CoV2/Flu/RSV, and the ID NOW COVID-19, respectively. For influenza A, it was 90.0% for the Idylla™ SARS-CoV2/Flu/RSV and 80.0% for the ID NOW Influenza. **Discussion**. All of the assays are suitable for testing patients with respiratory symptoms. False negatives should be considered, and the test should be repeated regarding the context.

## 1. Introduction

The burden of respiratory viruses such as SARS-CoV-2 and influenza is high in elderly and immunocompromised patients. The specific curative and prophylactic treatments available are more efficient if they are administered rapidly after the onset of the symptoms or the infectious contact [1,2]. Furthermore, the identification of infected patients is required, especially in hospital settings and community living spaces, in order to implement infection prevention and control measures to prevent outbreaks [3]. However, the SARS-CoV-2 pandemic has changed the epidemiology of respiratory viruses. After the almost-disappearance of “common” respiratory viruses, the activity of seasonal viruses has resumed [4,5,6]. Indeed, while some viruses were sparsely detected, the RSV 2020–2021 season was delayed by about 10 weeks. It is therefore likely patients with respiratory symptoms might be infected by SARS-CoV-2, influenza, or another respiratory virus such as RSV. 

The clinical presentation of respiratory viral infection is unspecific, and their etiological diagnostics require biological tests. Several assays based on antigen detection or molecular amplification, which is considered the gold standard for routine tests, are available, but their performances are heterogeneous. The Idylla™ SARS-CoV-2 (Biocartis, Mechelen, Belgium) and the Idylla™ SARS-CoV2/Flu/RSV (Biocartis) are fully integrated RT-PCR methods that allow the detection of SARS-CoV-2, and SARS-CoV-2, influenzas A and B, and RSV, respectively. While the performance of the Idylla™ SARS-CoV-2 was previously assessed, that of the Idylla™ SARS-CoV2/Flu/RSV had never been assessed to date [7,8]. The ID NOW COVID-19 (Abbott Rapid Diagnostic, Scarborough, ME, USA) and ID NOW Influenza A & B 2 (Abbott Rapid Diagnostic) are isothermal amplification methods that detect SARS-CoV-2 and Influenza A and B, respectively. All of the assays are user-friendly and provide results within 90 min and 15 min of processing for the Idylla™ and the ID NOW instruments, respectively. None of them were assessed in the epidemiological context of the co-circulation of SARS-CoV-2, influenza and RSV.

The aim of this study is to assess the performances of four commercial molecular assays for the detection of respiratory viruses in comparison to a reference multiplex RT-PCR.

## 2. Materials and Methods

### 2.1. Study Design

Four commercial fully integrated molecular assays—the ID NOW COVID-19, ID NOW Influenza A & B 2, The Idylla™ SARS-CoV-2, and the Idylla™ SARS-CoV2/Flu/RSV—were evaluated in comparison to a reference multiplex RT-PCR which is routinely used in our laboratory: the Alinity M RESP-4-Plex. The assays were performed on their respective and specific instruments: Alinity M (Abbott molecular, Des Plaines, IL, USA), ID NOW (Abbott Rapid Diagnostic), and Idylla (Biocartis, Mechelen, Belgium). The software of all of the instruments automatically interpreted amplification curves as positive, negative or uninterpretable. The respiratory viruses and the viral targets detected by each assay, as well as the volume of the sample required for each assay, are described in Table 1. 

The study was performed in two parts. As the Idylla assays have never been evaluated previously, we first assessed their analytical performances using a collection of clinical samples. Then, the performances of the four assays were assessed in a prospective study. All of the nasopharyngeal swabs (NPS) were sampled on universal transport media (UTM) using a Yocon virus sampling kit (Yocon biology technology company, Beijing, China) from symptomatic patients suspected of viral respiratory infections. All of the tests were performed on fresh NPS stored for a maximum of 16 h at +5 °C, except for those included for the assessment of cross-reactivity in the evaluation of the analytical performances of the Idylla assays (Table 2). Those samples were stored at −70 °C before testing. The screening of mutations E484K, E484Q, and L452R was performed using the IDTM SARS-CoV-2/VOC evolution Pentaplex (ID solutions, Grabels, France) for all of the SARS-CoV-2-positive samples. Statistical analyses were performed using R software [9].

### 2.2. Analytical Performances of the Idylla Assays

The performances of the Idylla™ SARS-CoV-2 and the Idylla™ SARS-CoV2/Flu/RSV were evaluated using a selection of NPS collected in December 2021: 15 SARS-CoV-2-positive fresh NPS (Ct range 15.1–37.3) tested with both assays, 11 influenza-positive (Ct range 15.4–38.1), and 11 RSV-positive (Ct range 17.2–36.4) fresh NPS tested with the Idylla COVID-Flu-RSV. As the Idylla assays amplified several viral targets of the same genes, the median Ct for each viral target was calculated.

In order to assess the cross-reactivity, twenty frozen NPS which were negative for SARS-CoV-2 but positive for another respiratory virus with the AllPlex RP1, RP2, and RP3 (Seegene, Seoul, South Korea) were selected. They were collected between April and October 2021. The number of NPS which were positive for each viral pathogen are listed in Table 3: four influenza (Idylla™ SARS-CoV-2 only), 3 parainfluenza, 1 metapneumovirus, 6 coronavirus, 3 human respiratory syncytial virus (RSV) (Idylla™ SARS-CoV-2 only), 1 human rhinovirus, 1 adenovirus, and 1 human enterovirus. 

The results of the Idylla™ SARS-CoV-2 and the Idylla™ SARS-CoV2/Flu/RSV were compared with those of the Alinity M RESP-4-Plex assay using the overall percentage of agreement, the positive percentage agreement, and a Kappa test.

### 2.3. Prospective Analysis

All of the adult patients suspected of respiratory viral infections admitted to the emergency department from 9 December 2021 to 21 December 2021 were enrolled. Socio-demographic and clinical data were prospectively recorded: date of birth, sex, date of first symptoms, nature of the symptoms, and SARS-CoV-2 vaccine status. Fresh NPS were processed using the Alinity M RESP-4-Plex (Abbott Molecular) as the reference RT-PCR in comparison to the ID Now COVID-19, the ID NOW Influenza A & B 2, Idylla™ SARS-CoV-2, the Idylla™ SARS-CoV2/Flu/RSV. The Alinity M RESP-4-Plex was repeated for all of the discrepant results, and performances were assessed after discrepant resolution. The sensitivity and specificity of each test was calculated for each assay and pathogen.

## 3. Results

### 3.1. Analytical Performances of the Idylla Assays

Of the 15 SARS-CoV-2-positive NPS, 13 (86.7%) and 12 (80.0%) were found positive with the Idylla™ SARS-CoV-2 and the Idylla™ SARS-CoV2/Flu/RSV, respectively (Figure 1). All of the samples which were negative with the Idylla assays had a Ct value greater than 30 with the Alinity M. Sample No. 925301 (Alinity M Ct value of 35.4) was found positive with the Idylla™ SARS-CoV-2 (one of five viral targets amplified with a Ct value of 39.9) but negative with the Idylla™ SARS-CoV2/Flu/RSV, although one of the three SARS-CoV-2 viral targets was amplified with a Ct value of 43.3. Sample No. 820201 (Alinity M Ct value of 37.3) was found negative with both assays, although a single target of the N gene was amplified in each assay with Ct values of 43.0 and 44.0 for the Idylla™ SARS-CoV-2 and the Idylla™ SARS-CoV2/Flu/RSV, respectively. No viral targets were amplified with both the Idylla™ SARS-CoV-2 and the Idylla™ SARS-CoV2/Flu/RSV for sample No. 641201, which displayed a Ct value of 33.0 with the Alinity M RESP-4-Plex. In comparison to the Alinity M SARS-CoV-2 target, the median Ct values of the N-gene and the ORF1b-gene were 2.5 and 6.1 cycles higher with the Idylla™ SARS-CoV-2, respectively. The Ct values were respectively 2.8 et 8.4 cycles higher with the Idylla™ SARS-CoV2/Flu/RSV. 

One of 11 (9.1%) RSV-positive NPS and three of 10 (30.0%) influenza A-positive NPS were not detected with the Idylla™ SARS-CoV2/Flu/RSV (Figure 1). The RSV-positive sample (No. 569901) displayed a Ct value of 36, while the three influenza A-positive NPS (No. 672701, 629501, and 008901) had Ct values of 32, 34, and 38 with the Alinity M RESP-4-Plex. In comparison to the Alinity M RESP-4-Plex, the median Ct value for positive samples was 6.7 cycles higher with the Idylla™ SARS-CoV2/Flu/RSV for both influenza A and RSV. The single influenza B sample was tested positive with the Idylla COVID-Flu-RSV. No cross-reactivity was identified with the NPS positive for another respiratory virus (Table 3).

The overall percentage agreements between the Idylla™ SARS-CoV2/Flu/RSV and the Alinity RESP-4-Plex were respectively 80.6%, 86.9%, and 95.8% for SARS-CoV-2, influenza A, and RSV. The positive percent agreements were 80.0%, 66.7%, and 90.9%, respectively, and the Kappa values were 0.88, 0.75, and 0.92. For the Idylla™ SARS-CoV-2, the overall percentage agreement and the positive percentage agreement were 94.3% and 86.7%, respectively, and the Kappa value was 0.88.

### 3.2. Prospective Analysis

Overall, 218 patients were enrolled in the prospective analysis. Their median age was 56.7 [interquartile range 40.4–77.3], and the male/female ratio was 0.93. The respective prevalence of SARS-CoV-2, influenza A, and RSV were 19.8% (*n* = 43), 4.8% (*n* = 10), and 3.2% (*n* = 7). No patient tested positive for influenza B. A single virus was detected in all of the patients, except for a SARS-CoV-2/RSV co-infection (patient No. 632901). Overall, the median delay from the onset of symptoms to NPS sampling was 3 days (interquartile range 2–7 days) for the patients infected with a respiratory virus. This timeframe was shorter for influenza A (2 days [1.3–3]), than RSV (3 days [3,4,5]) and SARS-CoV-2 (4 days [2–8.5])-positive patients. Among the SARS-CoV-2 positive patients, 28 (65.1%) were discharged on the day of admission, 10 (23.3%) were hospitalized in medical departments, and 5 (11.6%) were admitted to the intensive care unit. The median Ct of the RNAseP gene, assessed with the Idylla™ SARS-CoV2/Flu/RSV, was 30.8 (29.4–33.6). Eighteen (8.3%) patients had an RNAseP Ct value greater than 35.0, suggesting a suboptimal sampling according to the manufacturer’s instruction.

The numbers of tests providing no results due to processing error were one (0.5%) for the Idylla™ SARS-CoV-2 and the ID NOW influenza A & B 2, two (0.9%) for the Idylla™ SARS-CoV2/Flu/RSV, and three (1.4%) for the ID NOW COVID-19 assays. The Idylla™ SARS-CoV-2 and the Idylla™ SARS-CoV2/Flu/RSV were repeated using the same conditions: two of them provided a valid result, while the third provided no result with the Idylla™ SARS-CoV2/Flu/RSV, and was consequently considered invalid. Due to insufficient sample quantity, the ID NOW COVID-19 and the ID NOW Influenza A & B 2 could not be repeated. 

Eighteen NPS (8.3%) were considered invalid using the Idylla™ SARS-CoV2/Flu/RSV due to the low amplification of the human RNAseP gene (Ct value > 35.0). Of these, four (22.2%) were positive for a respiratory virus with the Alinity M RESP-4-Plex (Table 4): one RSV (No. 345101) and three SARS-CoV-2 (No. 959702, 811101, and 879801). At least one viral target was amplified for three of these samples with the Idylla™ SARS-CoV2/Flu/RSV: one RSV (No. 345101) and two SARS-CoV2 (No. 959702 and 879801). No viral target was amplified for the remaining one (No. 811101), while it tested positive with the ID NOW COVID-19, Idylla™ SARS-CoV-2 (the amplification of all viral targets with a Ct value > 39.8), and the Alinity M RESP-4-Plex (a Ct value of 31.5). 

No false positive was found with all of the assays for any of the respiratory viruses. The discrepancies between the reference test and the evaluated assays are listed in Table 5. The Ct value of a SARS-CoV-2 false-negative ranges from 31.2 to 36.4. All of the patients were aged less than 50 years. They had mild symptoms for 3 to 14 days, with three of them being symptomatic for at least 10 days before testing. All of the patients were discharged on the day of their admission. The single SARS-CoV-2 false negative of the Idylla™ SARS-CoV-2 was also found to be negative with the Idylla™ SARS-CoV2/Flu/RSV and the ID NOW COVID-19. Three other SARS-CoV-2-positive NPS were found negative with both the ID NOW COVID-19 and the Idylla COVID-Flu-RSV. Of note, despite the Idylla™ SARS-CoV2/Flu/RSV providing a negative result, at least one viral target was amplified for three NPS (No. 832301, 909701, and 909301) with a Ct value greater than 40.0. Except for sample No. 909301, the Ct of the RNAseP gene was greater than 33.0 for all of the samples. All of the patients hospitalized or admitted to the intensive care unit were identified with all three assays. The overall sensitivities for SARS-CoV-2 detection were 97.7%, 82.5%, and 86.3% for the Idylla™ SARS-CoV-2, the Idylla™ SARS-CoV2/Flu/RSV, and the ID NOW COVID-19, respectively.

There were two and one false negatives on the ID NOW Influenza A & B 2 and the Idylla™ SARS-CoV2/Flu/RSV for influenza A, respectively (Table 5). Sample No. 501001, which had a Ct value of 31 with the Alinity RESP-4-Plex, was negative with the ID NOW Influenza A & B 2 only. The patient was sampled 7 days after the onset of dyspnea. Sample No. 661701 (Alinity M Ct value of 36.0) was negative with both the ID NOW Influenza A & B 2 and the Idylla™ SARS-CoV2/Flu/RSV assays. All of the false negatives had a Ct value greater than 33.0 for the RNAseP gene. The sensitivity of the Idylla™ SARS-CoV2/Flu/RSV and the ID NOW Influenza were 90.0% and 80.0%, respectively, for influenza A. The single RSV false negative of the Idylla™ SARS-CoV2/Flu/RSV had a Ct value of 31.0 with the Alinity M. The sensivity of the Idylla™ SARS-CoV2/Flu/RSV was 83.3% for RSV.

## 4. Discussion

The main strength of our study was the prospective assessment of four molecular assays during the co-circulation of three respiratory viruses: SARS-CoV-2, influenza, and RSV. All of the assays were in agreement with the reference RT-PCR for NPS displaying a high viral load, i.e., a CT value lower than 30.0, whatever the viral target. The sensitivity of the Idylla™ SARS-CoV2 Test was the highest for SARS-CoV2 detection among the assays evaluated. No cross-reactivities or false positives were identified, confirming the high specificity of all of the assays. 

Influenza and RSV are seasonal viruses, with RSV mainly circulating between October and December, and Influenza mainly circulating between December and January. Since its emergence, SARS-CoV-2 has spread in a multiwave dynamic. Furthermore, Influenza and RSV activity strongly decreased after the beginning of the COVID-19 pandemic. All of these findings make the co-circulation of these viruses unlikely. Our study took place in December 2021, which was marked by the increased activity of SARS-CoV-2 and influenza and the end of the seasonal outbreak of RSV. This allowed us to assess four molecular assays in the singular context of viral co-circulation.

The evaluation of the Idylla™ assays using a collection of fresh NPS displaying various CT values shows that those displaying a CT value greater than 30.0 are likely not detected. Similar findings were previously reported for the ID NOW assays [10,11,12]. They reflect that the patients displaying a low viral load might be not detected using these assays. Notably, viable virus is rarely cultured at Ct values > 30.0 on or after 14 days of illness, suggesting that the probability of infectivity decreases with increasing Ct values [13,14,15,16]. Therefore, the CT value has been suggested as a parameter to take into account for the decision to discontinue isolation in hospitalized patients with COVID-19 [15,17,18]. Most of the SARS-CoV-2-positive patients included in our prospective study had a Ct-value below 30.0. These results suggest that a diagnostic assay should be evaluated both using a collection of clinical samples displaying various viral loads and prospectively on fresh clinical samples. However, a low CT value could be reported at the early or late stages of the disease, or in the case of suboptimal sampling. Therefore, while all of the evaluated tests appeared acceptable for routine use, a negative result should be interpreted with caution considering the context and clinical symptoms, and re-testing should be recommended in case of the persistence of symptoms. 

The sensitivity of a RT-PCR depends on pre-analytical issues such as the sampling, mainly for those displaying a Ct-value greater than 33.0, with a suboptimal sampling being associated with lower performances [19]. The Idylla™ SARS-CoV2/Flu/RSV amplified the RNAseP gene, a human housekeeping gene, allowing us to assess the quality of sampling. The main strategy of this assay is to avoid a false negative due to suboptimal sampling by providing an invalid result. In the present study, a significant number of NPS (8.3%) had low cellularity assessed with the Idylla™ SARS-CoV2/Flu/RSV. However, a viral target was amplified with a Ct value < 40.0 for three NPS, and they were found to be positive with the reference RT-PCR, as did the other evaluated assays. An inhibition during the analytical process of the Idylla COVID-Flu-RSV test can be excluded because the internal controls were amplified in the expected range. Therefore, we suggest that the algorithm of the Idylla™ SARS-CoV2/Flu/RSV be improved to provide a valid result for NPS reaching criteria of positivity whatever the amplification of the RNAseP gene. 

The time-to-result is challenging for clinical laboratories mainly for COVID-19 diagnosis, as infection prevention and control measures should be implemented rapidly and hospital departments are overwhelmed during the successive waves of the pandemic. The analytic process of the Idylla assays is 90 min regardless of the result. We suggest that it could be shortened for positive results by ending the analytic process immediately after the detection of positivity, as for the ID NOW [20,21]. This change would enhance the instrument’s throughput. However, the Idylla instrument can include up to eight modules performing as many simultaneous tests, which compensated for the longer time-to-result. 

The rate of invalid results remained low with all of the assays, which furthermore are user-friendly. Indeed, invalids could reach up to 35.5% with some assays [22]. Using the ID NOW instrument, invalids ranged from 0% to 7% [23,24,25,26], with this rate being lower when performed by laboratory-trained operators [22,23,25] in comparison to point-of-care use [24,26]. Consequently, the invalid rate of the Idylla™ instrument remains to be assessed if it is performed at the point of care. 

As the study was carried out in a single care center setting, the results might be specific to the population of this center and might not be applicable to another group of patients, such as children for example. A single influenza B-positive sample was included; the performances of the four assays, therefore, remain to be evaluated for this virus. All of the assays were performed on NPS sampled on UTM; using a fresh swab would avoid dilution and likely improve sensitivity.

## 5. Conclusions

In conclusion, we evaluated four molecular assays in the singular epidemiologic context of the co-circulation of SARS-CoV-2, influenza, and RSV. The sensitivity of the Idylla™ SARS-CoV-2 was higher than that of the Idylla™ SARS-CoV2/Flu/RSV and ID-NOW, but all of the assays were suitable for testing patients with respiratory symptoms. Nevertheless, due to lower performances for samples displaying a CT value higher than 30.0, a false negative should be considered and the test repeated regarding the context. We also suggest two improvements in the algorithm of the Idylla assays. First, while a main advantage of the Idylla™ SARS-CoV2/Flu/RSV is to assess the quality of the NPS, an NPS reaching the criteria of positivity but with low cellularity should be interpreted as positive rather than invalid. Then, in order to reduce the time-to-result, the analytic process could be ended after the detection of the positivity. 

## Figures and Tables

**Figure 1 jcm-11-03942-f001:**
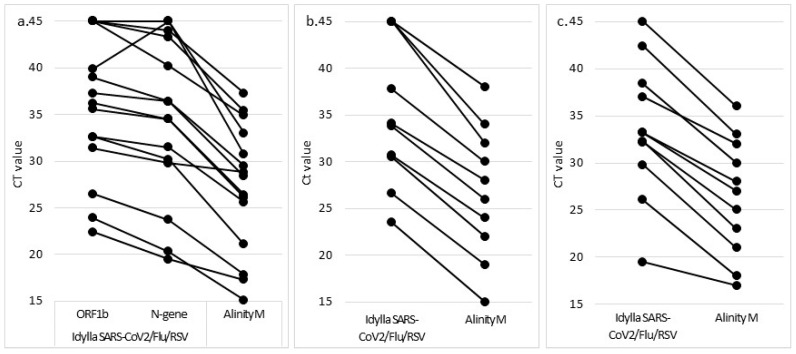
CT values of the viral targets obtained with the Idylla and the Alinity M assays during the study of the analytical performance. Idylla SARS-CoV2/Flu/RSV assay for SARS-CoV-2 (**a**), influenza A (**b**), and RSV (**c**) targets. Idylla SARS-CoV2 assay (**d**).

**Table 1 jcm-11-03942-t001:** Characteristics of the reference RT-PCR and evaluated assays.

	Method	Volume of Sample **	Viral Genes Amplified (Number of Fluorphore)			
			SARS-CoV-2	Influenza A	Influenza B	RSV
Alinity M RESP-4-Plex	RT-PCR	500 µL	ORF1b and N (1)	Matrix (1)	NS1 (1)	Matrix (1)
Idylla™ SARS-CoV-2	RT-PCR	200 µL	ORF1b (3) N (2)			
Idylla™ SARS-CoV2/Flu/RSV	RT-PCR	400 µL	ORF1b (1) N (2)	- *	-	-
ID NOW COVID-19	Isothermal amplification	200 µL	ORF1b (1)			
ID NOW influenza A & B 2	Isothermal amplification	200 µL		Matrix (1)	Matrix (1)	Fusion and Nucleocapsid (1)

* Pathogen detection not included in the assay. ** Recommended by the manufacturer.

**Table 2 jcm-11-03942-t002:** Number of NPS and tests performed in the analytical and the prospective studies.

	Conservation	Idylla™ SARS-CoV-2	Idylla™ SARS-CoV2/Flu/RSV	ID NOW COVID	ID NOW Influenza A & B 2
**Analytical**					
15 SARS-CoV-2	Fresh	X *	X		
11 influenza A	Fresh		X		
1 influenza B	Fresh		X		
1 RSV	Fresh		X		
20 Others viruses	Frozen	X	X		
**Prospective**					
218 NPS	Fresh	X	X	X	X

* The tests performed are marqued with an X.

**Table 3 jcm-11-03942-t003:** Panel of commonly found respiratory viruses in respiratory infections used for the evaluation of cross-reactivity.

Clinical NPS * with Known Viruses	Number Tested	
	Idylla™ SARS-CoV2/Flu/RSV	Idylla™ SARS-CoV-2
Influenza A	-	3
Influenza B	-	1
RSV	-	3
Parainfluenza 1	1	1
Parainfluenza 3	1	1
Parainfluenza 4	1	1
Coronavirus OC43	2	2
Coronavirus NL63	2	2
Coronavirus 229E	2	2
Human Metapneumovirus	1	1
Adenovirus	1	1
Enterovirus	1	1
Rhinovirus	1	1

* NPS: nasopharyngeal swab.

**Table 4 jcm-11-03942-t004:** Results for the four samples displaying a Ct value > 35.0 for the RNAseP gene but positive for a respiratory virus with the Alinity M.

Sample ID	Idylla™ SARS-CoV2/Flu/RSV (Ct Value)							Alinity M Target (Ct Value)	ID NOW COVID-19	ID NOW Influenza A & B 2	Idylla™ SARS-CoV-2
	ORF1b	N	N	Flu A	VRS	Flu B	RNAseP				
959702	29.1	27.2	27.3	n.d. *	n.d.	n.d.	37.2	COVID (17.6)	Positive	Negative	Positive
811101	n.d.	n.d.	n.d.	n.d.	n.d.	n.d.	37.6	COVID (31.5)	Positive	Negative	Positive
879801	31.1	28.9	29.1	n.d.	n.d.	n.d.	36.6	COVID (19.1)	Positive	Negative	Positive
345101	n.d.	n.d.	n.d.	n.d.	37.1	n.d.	36.5	RSV (24.0)	Negative	Negative	Negative

* n.d. not detected.

**Table 5 jcm-11-03942-t005:** Discrepancies between the Alinity M RESP-4-Plex and the evaluated assay for SARS-CoV-2 (a), influenza A (b), and RSV (c).

**(a)**													
**Sample ID**	**Sex**	**Age**	**Alinity M** **(Ct)**	**ID NOW COVID**	**Idylla™** **SARS-CoV-2**	**Idylla™ SARS-CoV2/Flu/RSV**			**Delay since Symptoms Onset**	**COVID-19 Vaccine**	**Symptoms**	**Oxygen Therapy**	**Outcome**
						**Qualitative Result**	**Targets Ct**	**RNAseP (Ct)**					
832301	F	21	31.2	Negative	Positive	Negative	ORF1b a = nd N b = 44.3 N c = 44.3	34,5	3	No	Abdominal pain	No	Discharge
930001	M	46	31.8	Negative	Negative	Negative	ORF1b a = nd N b = nd N c = nd	34.3	14	No	Flu-like syndrome	No	Discharge
882501	F	31	32.3	Negative	Positive	Negative	ORF1b a = nd N b = nd N c = nd	35.3	5	No	Flu-like syndrome	No	Discharge
909701	M	29	33.9	Negative	Positive	Negative	ORF1b a = nd N b = nd N c = 43.13	33.4	10	n.a.	Headhache hypoaesthesia of the right side	No	Discharge
909301	M	22	36.4	Negative	Positive	Negative	ORF1b a = nd N b = 41.15 N c = 43.29	30.2	14	No	Flu-like syndrome	No	Discharge
**(b)**													
**Sample ID**	**Sex**	**Age**	**Alinity M** **(Ct)**		**ID NOW Indluenza A & B 2**	**Idylla™ SARS-CoV2/Flu/RSV**			**Delay since** **Symptoms Onset**		**Symptoms**	**Oxygen Therapy**	**Outcome**
						**Flu A (Ct)**	**RNAseP**						
501001	M	68	31		Negative	Positive (39.7)	33.6		7		Dyspnea	4 L	Hospitalization
661701	M	39	36		Negative	Negative	33.6		1		Flu-like syndrome	No	Discharge
**(c)**													
**Sample ID**	**Sex**	**Age**	**Alinity M** **(Ct)**		**Idylla™ SARS-CoV2/Flu/RSV**				**Delay since** **Symptoms Onset**		**Symptoms**	**Oxygen Therapy**	**Outcome**
					**RSV**	**RNAseP E**							
842101	F	81	31		Negative	33.4			n.a.		Dyspnea	6 L	Hospitalization

## Data Availability

The data presented in this study are available on request from the corresponding author.

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
