# Peer review of "Evaluation of Four Fully Integrated Molecular Assays for the Detection of Respiratory Viruses during the Co-Circulation of SARS-CoV-2, Influenza and RSV"

_jcm, 2022, doi:10.3390/jcm11143942_

Round 1
Reviewer 1 Report
In this study, the authors have tested the performances of 4 integrated molecular assays. They assessed the sensitivity and specificity using a collection of nasopharyngeal swabs, as well as in a prospective study. Overall, they found 4 assays were 100% specific, but much less sensitive compared to a reference assay
I feel that the publication of these results may come a bit late concerning the timing of the pandemic. In addition, the assays seem much less sensitive compared to the reference assay.
I think the experiments using standards with known concentrations should be performed to assess the sensitivity of the assays.
Other comments.
1) Please clarify why Alinity M RESP-4 Plex was chosen as the reference assay.
2) Line 69, please define NPS. And "NPS" in Line 85 is the second time in the text.
3) Please describe which sampling kit was used to collect NPS sample.
4) Please specify how much volume of sample was used for the assays.
5) Please describe how the threshold is defined for each assay.
6) Do these essays need to be performed on a real-time PCR machine? if yes, which machine did they use? I guess no additional machine is needed for these assays, right?
7) It is not clear how these commercial kits determine "positive" and "negative", please describe this in the methods.
8) If it is possible, please provide more information about the sequence of primers and probes, and also real-time PCR, is it TaqMan or SybGreen?
9) I think it is better to use figures to visualise the data. As Fig. 3b in this paper https://www.nature.com/articles/s41564-020-0761-6
10) Line 129 -131, please add statistical analysis.
11) Line 150 - 153, please check the data in the table, they are not consistent with the description in the results.
Author Response
Dear Reviewer,
Thank you for your comments that greatly improved the manuscript. We made several changes that appear in red in the manuscript.
Best Regards
Eric Farfour
Reviewer 1
In this study, the authors have tested the performances of 4 integrated molecular assays. They assessed the sensitivity and specificity using a collection of nasopharyngeal swabs, as well as in a prospective study. Overall, they found 4 assays were 100% specific, but much less sensitive compared to a reference assay
I feel that the publication of these results may come a bit late concerning the timing of the pandemic. In addition, the assays seem much less sensitive compared to the reference assay.
I think the experiments using standards with known concentrations should be performed to assess the sensitivity of the assays.
Answer: We assessed the performances of 4 molecular assays in the context of changes in the epidemiology of respiratory viruses. The SARS-CoV-2 pandemic has begun more than 2 years ago, and SARS-CoV-2 circulation is nearly endemic in several parts of the world. Furthermore, other respiratory viruses such as influenza and RSV, that have almost disappeared during the first months of the pandemic, were subsequently detected. As for winter 2021-2022, circulation of SARS-CoV-2, influenza, and RSV during the winter is likely to occur in the next years. As we performed a "real-life" study, we did not use standards with known concentrations of viruses.
Other comments.
1) Please clarify why Alinity M RESP-4 Plex was chosen as the reference assay.
Answer: The Alinity M RESP-4-Plex is routinely used in our clinical laboratory. It was chosen as it could detect all 4 respiratory viruses with a random-access workflow providing rapid results.
- Previously: Four commercial fully integrated molecular assays were evaluated in comparison to a reference multiplex RT-PCR, the Alinity M RESP-4-Plex
- Revised: Four commercial fully integrated molecular assays were evaluated in comparison to a reference multiplex RT-PCR routinely used in our laboratory, the Alinity M RESP-4-Plex
2) Line 69, please define NPS. And "NPS" in Line 85 is the second time in the text.
Answer: NPS is defined line 69 in the revised manuscript
- Previously: All the NPS had been sampled on universal transport media
- Revised: All the nasopharyngeal swabs (NPS) had been sampled on universal transport media
3) Please describe which sampling kit was used to collect NPS sample.
Answer: The sampling kit used is added in the revised manuscript
- Previously: All the nasopharyngeal swabs (NPS) had been sampled on universal transport media (UTM)
- Revised: All the nasopharyngeal swabs (NPS) had been sampled on universal transport media (UTM) using Yocon virus sampling kit (Yocon biology technology company, Beijing, China)
4) Please specify how much volume of sample was used for the assays.
Answer: The volume of sample recommended by the suppliers were used for the assays:
- Previously: The respiratory viruses and the viral targets detected by each assay are described in Table 1.
- Revised: The respiratory viruses and the viral targets detected by each assay, as well as the volume of sample required for each assay, are described in Table 1.
The volume of sampled required for each assay is added in table 1.
|
|
Method |
Recommended volume of sample |
Viral genes amplified (number of fluorphore) |
|||
|
|
SARS-CoV-2 |
Influenza A |
Influenza B |
RSV |
||
|
Alinity M RESP-4-Plex |
RT-PCR |
500 µL |
ORF1b and N (1) |
Matrix (1) |
NS1 (1) |
Matrix (1) |
|
Idylla™ SARS-CoV-2 |
RT-PCR |
200 µL |
ORF1b (3) N (2) |
|
|
|
|
Idylla™ SARS-CoV2/Flu/RSV |
RT-PCR |
400 µL |
ORF1b (1) N (2) |
-* |
- |
- |
|
ID NOW COVID-19 |
Isothermal amplification |
200 µL |
ORF1b (1) |
|
|
|
|
ID NOW influenza A & B 2 |
Isothermal amplification |
200 µL |
|
Matrix (1) |
Matrix (1) |
Fusion and Nucleocapsid (1) |
5) Please describe how the threshold is defined for each assay.
Answer: The threshold is defined by the manufacturer. Unfortunately, these data are not publicly available.
6) Do these essays need to be performed on a real-time PCR machine? if yes, which machine did they use? I guess no additional machine is needed for these assays, right?
Answer: The assays were performed on their respective and specific instruments. This data is added in the manuscript
- Revised (line 62): All assays were performed on their respective and specific instrument: Alinity M (Abbott molecular), ID NOW (Abbott Rapid Diagnostic), and Idylla (Biocartis).
7) It is not clear how these commercial kits determine "positive" and "negative", please describe this in the methods.
Answer: The amplification curves are interpreted by the software of each instrument. The algorithm is not publicly available.
- Revised (line 64): The software of all instruments automatically interpreted amplification curves as positive, negative or uninterpretable.
8) If it is possible, please provide more information about the sequence of primers and probes, and also real-time PCR, is it TaqMan or SybGreen?
Answer: The sequence of primers and probes are not available publicly for these commercial assays.
The methods are either derived of TaqMan for Alinity M and Idylla or based on isothermal amplification for the ID NOW. Please see table 1.
9) I think it is better to use figures to visualise the data. As Fig. 3b in this paper https://www.nature.com/articles/s41564-020-0761-6
Answer: Tables 4 and 5 were removed from the manuscript and replaced by a figure (Figure 1).
10) Line 129 -131, please add statistical analysis.
Answer: Figure 1. greatly improve the visualization of the CT value. No statistical analysis were performed in the paper https://www.nature.com/articles/s41564-020-0761-6 . Regarding figure 1 and the median differences of CT value, the contribution of statistical analysis would be low.
11) Line 150 - 153, please check the data in the table, they are not consistent with the description in the results.
Answer: There was a mistake in table 5b and 5c. Both tables are removed from the manuscript (See comment 9) and replaced by a figure (figure 1).
Reviewer 2 Report
In this very interesting study, the authors present an evaluation of several respiratory virus detection kits/methods. The manuscript is clear and concise, but there a few specific points that need to be clarified before publication.
Line 67: The phrasing here is awkward, suggest maybe "As the assays have never been evaluated..."
Line 92: October is misspelled
Consider moving Tables 4 and 5 to supplemental information. The CT values are interesting and important, but too much to include in the text itself.
In the discussion section, it would be helpful to include a limitations section.
Author Response
Dear Reviewer,
Thank you for your comments that greatly improved the manuscript. We made several changes that appear in red in the manuscript.
Best Regards
Eric Farfour
Reviewer 2.
In this very interesting study, the authors present an evaluation of several respiratory virus detection kits/methods. The manuscript is clear and concise, but there a few specific points that need to be clarified before publication.
Line 67: The phrasing here is awkward, suggest maybe "As the assays have never been evaluated..."
Answer: The sentence is corrected in the revised manuscript:
- Previously: As the Idylla assays were never evaluated previously,
- Revised: As the Idylla assays have never been evaluated previously,
Line 92: October is misspelled
Answer: The sentence is corrected in the revised manuscript:
- Previously: They were collected between April and Octobre 2021
- Revised: They were collected between April and October 2021
Consider moving Tables 4 and 5 to supplemental information. The CT values are interesting and important, but too much to include in the text itself.
Answer: Tables 4 and 5 were removed from the manuscript and as suggested by the reviewer 1 replaced by a figure (Figure 1.)
In the discussion section, it would be helpful to include a limitations section.
Answer: A limitations section is added in the revised manuscript:
- Revised: As the study was carried out in a single care center setting, the results might be specific to the population of this center and not be applicable to another group of patients such as children for example. A single influenza B-positive sample was included; the performances of the 4 assays, therefore, remain to be evaluated for this virus. All assays were performed on NPS sampled on UTM, using fresh swab would avoid dilution and likely improve sensitivity.
Round 2
Reviewer 1 Report
Thanks very much for sending back the revised version.